# Progress in Glioma Stem Cell Research

**DOI:** 10.3390/cancers16010102

**Published:** 2023-12-24

**Authors:** Vanajothi Ramar, Shanchun Guo, BreAnna Hudson, Mingli Liu

**Affiliations:** 1Department of Microbiology, Biochemistry & Immunology, Morehouse School of Medicine, Atlanta, GA 30310, USA; vramar@msm.edu (V.R.); bhudson@msm.edu (B.H.); 2Department of Chemistry, Xavier University, 1 Drexel Dr., New Orleans, LA 70125, USA; sguo@xula.edu

**Keywords:** glioblastoma, molecular mechanism, NF-κB, drug targets, GSCs, TMZ

## Abstract

**Simple Summary:**

Glioblastoma stands as the most aggressive form of adult brain cancer, often resulting in a median survival of 14.6 months even following surgical intervention coupled with chemotherapy and radiotherapy. This review places emphasis on elucidating the current understanding of glioblastoma stem cell (GSC) biology, the intricacies of GSCs’ molecular mechanisms, and their correlations with diverse signaling pathways. Notably, this includes exploration of pivotal pathways such as epidermal growth factor receptor, PI3K/AKT/mTOR, HGFR/c-MET, NF-αB, Wnt, Notch, and STAT3. Moreover, this review investigates the phenomenon of metabolic reprogramming observed within GSCs. Additionally, it delves into potential therapeutic avenues targeting GSCs, presenting current inhibitors along with their respective modes of action on GSC targets.

**Abstract:**

Glioblastoma multiforme (GBM) represents a diverse spectrum of primary tumors notorious for their resistance to established therapeutic modalities. Despite aggressive interventions like surgery, radiation, and chemotherapy, these tumors, due to factors such as the blood–brain barrier, tumor heterogeneity, glioma stem cells (GSCs), drug efflux pumps, and DNA damage repair mechanisms, persist beyond complete isolation, resulting in dismal outcomes for glioma patients. Presently, the standard initial approach comprises surgical excision followed by concurrent chemotherapy, where temozolomide (TMZ) serves as the foremost option in managing GBM patients. Subsequent adjuvant chemotherapy follows this regimen. Emerging therapeutic approaches encompass immunotherapy, including checkpoint inhibitors, and targeted treatments, such as bevacizumab, aiming to exploit vulnerabilities within GBM cells. Nevertheless, there exists a pressing imperative to devise innovative strategies for both diagnosing and treating GBM. This review emphasizes the current knowledge of GSC biology, molecular mechanisms, and associations with various signals and/or pathways, such as the epidermal growth factor receptor, PI3K/AKT/mTOR, HGFR/c-MET, NF-κB, Wnt, Notch, and STAT3 pathways. Metabolic reprogramming in GSCs has also been reported with the prominent activation of the glycolytic pathway, comprising aldehyde dehydrogenase family genes. We also discuss potential therapeutic approaches to GSC targets and currently used inhibitors, as well as their mode of action on GSC targets.

## 1. Background

Glioblastoma is one of the most overwhelming primary brain tumors. The standard-of-care treatment involves maximal surgical resection followed by chemotherapy and radiotherapy. Temozolomide (TMZ) is one of the currently used drugs for the treatment of glioblastoma. Despite treatment with surgical resection and chemotherapy, patients only have a limited overall survival rate (15–19 months) due to factors such as the blood–brain barrier, tumor heterogeneity, glioma stem cells (GSCs), drug efflux pumps, and DNA damage repair mechanisms [1,2,3]. Extensive research has shown that GSCs play an important role in glioblastoma and are associated with increased resistance to chemotherapy and radiotherapy, implying that GSCs have a significant impact on successful treatment outcomes and contribute significantly to the high recurrence rates seen in glioblastoma [4,5,6]. As a result, GSCs are considered prospective therapeutic targets for glioblastoma therapy, and eliminating GSCs is critical in treating glioblastoma [7,8]. GSC targeting is primarily focused on the direct ablation of GSC cell surface markers and certain pathways involved in GSC stemness maintenance. Though it has evolved, another approach for selectively targeting GSCs is to change their ability and disturb their microenvironment, including their reliance on angiogenesis and immunological invasive characteristics [9,10]. 

The old names for glioblastoma multiforme (GBM) include various types of histologic features, such as pleomorphic cell GBM (26%), gemistocytic GBM (25%) [11], small-cell GBM (27%), gliosarcoma (2%), GBM with oligodendroglioma component (15%) [12], and mixed variants composed of more than one histologic pattern. The oligodendroglioma components of GBM show both astrocytic and oligodendroglia areas; some studies reported that they may be associated with a more favorable prognosis [13]. The World Health Organization (WHO) classified GBM with an oligodendroglioma component based on necrosis in the former tumor and distinguished it from anaplastic oligoastrocytoma. Small-cell GBMs are highly cellular and cytologically monotonous neoplasms with recurrent mitoses [14]. Gliosarcoma is a biphasic tumor showing a malignant mesenchyme tumor area and conventional GBM, whereas giant-cell GBM has extremely large multinucleated and highly pleomorphic cells [15].

More than 50% of glioma/GBMs are a common and aggressive type of brain tumor; hence, the complete resection of the tumor is not possible owing to its highly invasive properties. However, a number of combination therapies with radiotherapy and chemotherapy along with surgery can prolong the patient’s survival; however, the prognosis is still poor with a median survival of only 14.6 months [16,17]. 

The mechanism of chemoresistance is one of the crucial factors that leads to poor survival in GBM patients. Hence, finding novel strategies and developing potential and effective therapeutic molecules against GBM is essential to overcome chemoresistance [18]. Multiple mechanisms are involved in the chemoresistance of GBM, that is, overexpression of drug efflux transporter pumps such as P-glycoprotein, cancer stem cells, augmented DNA repair mechanisms, and dysregulation of apoptosis; this is considered to be an important factor responsible for drug resistance in GBM [19]. These factors highly influence the remarkable changes in tumor cells that lead to chemoresistance, but recently, new evidence has shown that the tumor microenvironment also plays a key role in chemosensitivity [20,21]. In addition to tumor cells, tumor lesions also contain a variety of stromal cells, including inflammatory blood cells, which infiltrate tumors, and endothelial cells (EC), which form blood vessels [22]. Additionally, these stromal cells with unique properties of cancer biology are present in the tumor microenvironment. Since the tumor microenvironment has been investigated for many years, the concept of “seed and soil” explains the phenomenon of distinct recurrent cancer metastasis [23]. In GBM, this tumor microenvironment is associated with tumor progression; hence, it has been well studied and may be considered as a potential drug target in anticancer treatment [24,25]. The initial way to identify GSCs is to select for markers of neural and hematopoietic stem cells, including CD133 (prominin) [26]. Approximately 100 CD133+ glioma cells were shown to be sufficient in a groundbreaking work to create xenografted tumors that accurately reflected the heterogeneity of the original tumor, while CD133- cells were efficiently depleted of their ability to cause tumors. According to the following findings, there has been confirmation and refutation of CD133’s significance as a GSC marker. Numerous samples, and even the same tumor types with CD133+ GSCs, showed tumorigenic CD133- cells [27,28,29,30]. A lack of technical consensus can only partially explain these controversial observations. Alternatively, the function of CD133 may differ throughout glioma molecular subtypes. Numerous studies demonstrated that while CD133-GSCs are strongly linked to mesenchymal subtypes, gliomas produced by CD133+ cells have transcription profiles that reflect proneural subtypes. According to Heidi Philips and colleagues’ work, CD133-GSCs drive a more complex hierarchy that results in CD133+ intermediate progenitors, which in turn give birth to CD133-differentiated progenies [31,32]. 

In addition, several markers that are associated with GBM markers, such as CD15, LICAM, A2B5, integrin, and α-6, are identified as other markers. Robust approaches for identifying and enriching GSCs are critical to the field [28,33]. However, a universal marker cannot encompass the genetically and phenotypically diverse GSCs. The recent advances and high-throughput technologies in genomic and epigenomic markers selective for subtypes of GSCs may be anticipated. This review emphasizes the current knowledge of GSC biology, including the molecular mechanism and association of various pathways, and GSCs’ role in GBM. We also discuss potential therapeutic approaches to GSC targets and currently used inhibitors, as well as their mode of action on GSC targets [34]. 

## 2. Molecular Mechanism of Glioma/Glioblastoma Cells

According to the clinical history of GBM patients, the histological features and genetic alteration are discerned and classified into two major types: primary or de novo GBM and secondary GBM. The primary variety is the most aggressive, known as IDH wild-type, and it typically arises in older patients with no history of lower-grade malignancies. Secondary GBMs that developed from prior low-grade gliomas (both grades II and II) with mutant IDH1/2 were associated with decreased tumor aggressiveness [35]. The amplification and/or mutation of the epidermal growth factor receptor (EGFR) gene (7p11.2) is one of the genetic alterations in GBM that occurs in 36% to 60% of primary GBMs [36]. Another typical genetic change is the 801-bp inframe deletion of exons 2 to 7 in mutant type variation 3 EGFRvIII [37]. Because it activates the EGFR-phosphatidylinostital3-kinase (PI3K) pathway and causes EGFR overexpression in primary GBMs with EGFR amplification, this change results in the constitutively active expression and higher cellular proliferation and survival rate of mutant cells [37]. 

The cyclin-dependent kinase inhibitor 2A (CDKN2A) gene’s homozygous deletion encodes the p16NK4a and p14ARF tumor suppressors, which are more frequently observed in both primary and secondary GBM [38]. Members of the cyclin-D-dependent protein kinases CDK4 and CDK6 phosphorylate the retinoblastoma protein 1 (RB1), which stimulates the transcription factor E2F expression and supports cell-cycle progression from G1 to S phase. Approximately 80% of the changes in primary GBM are connected with the CDKN2A/p16-CDK4/6-RB pathway, with CDKN2A gene deletion or mutation, CDK4 amplification, and RB1 mutation or deletion being the most common causes [39]. 

A total of 70% of initial GBMs are caused by chromosome 10 loss of heterozygosity (LOH). The deletion of chromosomal region 10q23–24 is the most prevalent (25%) change, which is where PTEN is located [40]. Primary GBM frequently exhibits chromosome 10 deletion in conjunction with EGFR signaling, implying that the interplay of EGFR signaling and the suppressor gene located on chromosome 10 is responsible for the aggressive nature of GBM. The isocitrate dehydrogenase (IDH) gene mutation is another major genetic modification in gliomas that is associated with an enhanced DNA hypermethylation profile; patients with mutated IDH1 and two gliomas have a better prognosis than patients with wild-type gliomas [41]. Parsons et al. [42] showed the existence of a recurrent point mutation in the active region of IDH1, and this IDH1 mutation is tightly related with TP53 mutation with del(1p)/del(19q), which may indicate that an early event in IDH mutation is more common in secondary GBM (80%) compared to initial GBM (5%). Previous research found that the O6-methylguanine DNA methyltransferase (MGMT) gene (chromosome 10q26) is frequently silenced by promoter hypermethylation in connection or not with monosomy of 10/del(10q). Indeed, a link has been established between MGMT-promoter methylation and improved patient outcomes with TMZ-based chemoradiation, owing to MGMT’s ability to reserve methylation at the O6 position of guanine and thus neutralize the cytotoxic effects of alkylating agents such as TMZ. MGMT methylation was just identified as a strong, clinically relevant component in GBM, and mandatory testing for this biomarker in standard practice is extremely contentious [43]. The MGMT promoter is methylated in approximately 50% of initially diagnosed GBM and 73% of secondary GBM. Less than 2% of GBM migrates beyond the brain, with most GBM cells infiltrating into healthy brain cells via the perivascular space around the blood vessels and the brain parenchyma space that contains both glial and neuron cells. A number of factors are needed for the penetration of glioma cells: ion channels, energy metabolism, cytoskeleton, remodeling of the extracellular matrix (ECM), neurotransmitters, proteases, and cell adhesion. The brain’s ECM is significantly involved in several cellular processes including migration and invasion of glioma cells that are associated with an altered microenvironmental composition. In addition, the intracellular Ca^2+^ signaling via inositol 1, 4, 5-triphophate receptor (IP3Rs), transient receptor potential (TRP) channels, store-operated channels (SOCs), and voltage-gated Ca^2+^ channels contributes to the motility of glioma cells [44,45]. 

In the infiltration of glioma in the brain area using cell–ECM interaction and associated dynamics, the proteins such as tenascins, proteoglycans, and hyaluronans play a major role in the brain ECM during invasion and are associated with binding partners like CD44, RHAMM, and integrins, which also have vital roles. The molecules of the ECM, such as reelin, laminins, tanascin-R, tenascin-C, and heparin-binding growth associated molecules, are responsible for synaptic plasticity and neuronal activity. The interaction of various components of ECM with other cell surface recognition molecules, ion channels, and receptors affects the synaptic plasticity by regulating the Ca^2+^ influx in endocytic zones and ECM remodeling [46]. 

## 3. Pathways Associated with Glioma and Glioblastoma

### 3.1. Epidermal Growth Factor Receptor (EGFR) Pathway

The EGFR signaling pathway is the most common and crucial pathway highly involved in cell survival, proliferation, growth, and differentiation in mammals. It has been investigated in depth using both experimental and computational approaches. Further research is essential to gain insights into the various intracellular dynamics and expand the signaling system scope [47]. Recently, a consortium was formed that is specifically focused on the receptor tyrosine kinase signaling system. The comprehensive map of molecular interaction may serve as a useful reference and help the research in EGFR signaling. EGFR is mapped in chromosome 7 short arm q22, spanning 110 kb of DNA and divided into 28 exons. In normal cells, the expression of EGFR is estimated to be from 40,000 to 100,000, whereas in tumor cells, more than 106 receptors per cell are overexpressed [48]. EFG significantly regulates their receptors to induce EGFR RNA expression by regulating the expression of ETF (EGFR-specific transcription factor). The other proteins, such as E1A, Sp1, and AP2, highly modulate the EGFR promoter; the interaction of DNA topoisomerase I and c-JUN also regulates EGFR gene expression. The EGFR gene has 28 exons and encodes the transmembrane protein receptor composed of 464 amino acids. Exons 5–7 encode the ligand-binding domain, exons 18–24 encode the tyrosine kinase domain, and exons 25–28 encode the autophosphorylation site [49,50].

However, EGFR is the most crucial drug target in tumor therapies; its mutations present an organ site asymmetry based on the origin of the organ. The mutation occurs in the kinase domain in other tumors; in gliomas, the mutation and deletion occur in ligand-binding ectodomain (ECD). These tissue-specific features lead the type II tyrosine kinase inhibitors with high sensitivity for inactive symmetric KD dimer when administered in a GBM mutation [51].

Except for hematopoietic cells, all cell types contain members of the ErbB family. The EGFR family of genes is required for normal vertebrate development. In mice, for example, a null mutation in the ErbB gene causes embryonic or perinatal death. The specific phenotype is determined by the mouse’s genetic background as well as the ErbB family members that were knocked out. The mortality of EGFR null mice has been linked to defects in organs such as the brain and skin, in the gastrointestinal system, and in stem cell renewal. The ERBB2 null mice die from cardiac trabeculae dysfunction, which exhibits malformation in motor nerves and sensory ganglia [50,52]. Several studies reported the overexpression of EGFR in more than 60% of primary GBM, whereas only 10% of secondary GBM is associated with a more aggressive GBM phenotype. Overexpression of EGFR causes an abnormal induction of EGFR activation in GBM as well as increased autocrine expression of related ligands. In 57% of GBMok, EGFR gene amplification and mutation increase EGFR activation and expression [51]. 

From a subtype standpoint, classical GBM is associated with 95% EGFR amplification, whereas mesenchymal, neural, and proneural GBMs have lower rates of EGFR amplification of 29, 67, and 17%, respectively. EGFR mutations are seen in around one-third of all classical tumors and are frequently found in mesenchymal, proneural, and neural GBMs as well. Extracellular domain EGFR mutations are the most typically found in GBM [18]. The alteration EGFRvIII is a common type mutation in GBM that results from an inframe deletion of 801 bp in the DNA sequence encoding the extracellular domain; it is not lost in normal tissues. As a result, this may serve as an effective and appealing pharmacological target for therapeutic intervention. According to several studies, the EGFRvIII is strongly expressed in 50% of GBM and amplifies the wild-type EGFR. According to TCGA data, EGFRvIII is typically found in classical cancers where EGFR amplification is most prominent [53,54,55,56].

The interaction of EGFR with tyrosine kinase activates PI3K and leads the phosphorylated phosphatidylinositol 4,5 biphosphate (PIP2) to form phosphatidylinositol 3,4,5-triphospahtes (PIP3). PIP3 activation lowers AKT activation while increasing the expression of mTOR complex 1 and 2 [57]. Activation of mTORC1 increases cellular development by enhancing the anabolic process of other macromolecules such as proteins, lipids, and organelle formation while decreasing the catabolic response, whereas activation of mTORC2 improves cell survival and proliferation. Several proteins are involved in regulating this pathway, for instance, the PTEN, an inhibitor of PIP2 phosphorylation, highly influences the regulation of this pathway [58]. 

The wild-type EGFR activates the mTORC1 pathway extensively, but the mutated EGFRvIII (deletion mutant) activates the mTORC2 pathway and contributes to EGFR-targeted therapy resistance. The AKT phosphorylation promoted by EGFRvIII efficiently decreases the production of P27KIP1, a cell-cycle regulator, and inhibits the G1- to S-phase transition. According to in vitro findings, EGFRvIII mutations in glioma cell lines greatly boosted the expression of the aberrant spindlelike microcephaly-associated protein (ASPM protein) and MMP-13, both of which promote neural stem cell regeneration and tumor invasiveness [59,60,61,62,63]. Similarly, increased production of VEGF and interleukin-8 (IL-8) via the nuclear factor kappa-light chain enhancer of activated B cells (NFκB) pathway promotes angiogenesis, which leads to GBM progression. Despite this, both the active and suppressed levels of EGFRΔIII influence the evading internalization and downregulation that results from inefficient dimerization [64]. In the wild type, the rapid degradation of EGFR is observed following acute stimulation with ligand. The low-level expression of EGFR EGFRΔIII effectively increases cell survival in GBM via selective augmentation of several mitogenic factors, including Akt. The EGFR signaling network thus acts as an effective drug target for therapeutic application. Tumor cells can become dependent on activated oncogenes, which is known as oncogene addiction. Several studies demonstrated a low rate of response to EGFRvIII inhibitors in GBM in the early stages. PTEN coexpression in GBM patients seems to make them more receptive to anti-EGFR therapy with erlotinib [62]. However, the association of EGFRvIII expression with PTEN was not predictive of increased survival in erlotinib-treated patients. According to the research, the more complicated molecular signature of this protein expression with individual tumors has to be found for effective GBM treatment. Furthermore, EGFR mutations involving tyrosine kinase identified in lung cancer may be more susceptible to TKI than GBM [65]. 

Several mechanisms are involved in chemotherapy resistance, and the EGFRvIII mutation also regulates the apoptotic proteins via Bcl-XL upregulation. This is one of the mechanisms that significantly influence the GBM recurrence in tumors with suppressed EGFRvIII expression [37]. 

A number of preclinical studies also reported a substantial proapoptotic effect with Bcl-XL antisense RNA and with a combination of EGFR-targeted tyrosine kinase inhibitors (TKIs), which might prove the efficiency of EGFR therapy. The study carried out by Bonavia and coworkers reported the expression of EGFRvIII and its impact on IL-6 [66,67]. The high expression of EGFRvIII influenced by IL-6 promoted the proliferation of tumor cells in vitro, and IL-6 significantly influenced the expression of Bcl-XL and STAT3 in another tumor. Several cellular subtypes are involved in GBM; among them, EGFR plays an important role with other receptors. Different receptor tyrosine kinases are also altered in GBM, leading to downstream activation of some other proteins that are activated by EGFR and that are highly influenced by survival regardless of EGFR inhibitors [68]. Another protein, PTEN, plays a key role in the EGFR inhibitor pathway because it has been shown to contribute to resistance in EGFR-targeted therapies. In PTEN-deficient cell lines, EGFR is activated by downstream signaling permanently and manipulates the autophagic pathway by preventing TKI-induced caspase activation via accumulation of heat-shock protein αB-crystallin. The proapoptotic effects also remarkably increased after treatment with erlotinip and mTOR inhibitors, inhibiting the autophagy-specific events [69]. The EGFR family complex system is also engaged in growth factor cellular signaling; phosphorylation of EGFR at the plasma membrane leads to the recruitment of several proteins via the binding of Src homology 2 (SH2) and phosphotyrosine-binding (PTB) domains to the receptor’s phosphotyrosine motifs [70]. The EGFR signaling complex initiates a number of signaling cascades that are crucial in tumor proliferation, motility, differentiation, and survival (Figure 1). Surprisingly, comparable substrates are activated with varying degrees of strength downstream of EGFR and EGFRvIII. The phosphoinositide 3-kinase (PI3K), signal transducer and activator of transcription number, Src family kinases, and mitogen-activated protein kinase (MAPK) pathways also play a crucial role in the EGFR downstream mechanism [71]. The heterodimer form of PI3Ks significantly triggers the adaptor proteins and RTKs via regulatory subunits such as p50a, p55a, and p85a or PIK3R1. The interaction of p85a with EGFR via ErB3 heterodimerization or phosphorylation of EGFR by c-Src results in conformational changes in p85a and the release of inhibition of PI3K catalytic subunit p110. As a result, it is found in the plasma membrane, where it catalyzes the synthesis of phosphatidylinositol 3,4,5-triphosphate (PIP3) by phosphorylation of PIP2 [72].

PIP3 production serves as an important activator of Akt, which then phosphorylates or inhibits a variety of proteins required for cellular metabolism, motility, and protein synthesis. When Akt is activated, it phosphosphorylates the Bcl family member Bad when it fails to inhibit the survival protein Bcl-xL, preventing apoptosis [73]. The activation of PI3K was also altered by a single mutation found in 15% of GBM. These mutations are most common in the adaptor-binding domain and less frequent in the catalytic subunit’s C2 helical kinase domains (PIK3CA). The aberrant PI3K activation leads to the activation of Akt, which is observed in 85% of GBM samples [57]. PI3K signaling is negatively regulated via proteins including PTEN. The loss of PTEN expression influences the PI3K: PTEN balance, resulting in increased Akt activation and controlled cell growth. The inhibitors like the rapamycin analogs everolimus and temsirolimus inhibit the mammalian target of rapamycin complex 1 (mTORC1) and are regulatorily approved for treating advanced renal cell carcinoma and GBM [74]. Research indicates that the utilization of rapamycin analogs in clinical settings has resulted in rare and brief reactions in GBM. Enzastaurin is the first targeted therapy that effectively inhibits the PKC/PI3K/AKT and is used for treating GBM evaluated in phase III clinical trials [75]. When EGFR activation is highly expressed, growth factor receptor-bound protein 2 (Grb2) binds to EGFR either directly through Y1068 and Y1086 or indirectly through SHC binding Y1173 and Y1143. This triggers the MAPK signaling cascade. Additionally, Grb2 contains two SH3 domains that enable interaction with protein-rich regions, such as those found in Son of Seven Less (SOS). A guanine nucleotide exchange factor called SOS is recruited to the plasma membrane as a result of the Grb2/Shc/EGFR interaction. SOS effectively facilitates the conversion of Ras-GDP to the active Ras-GTP. Ras then triggers the serine-threonine protein kinase Raf, which in turn triggers MEK1/2, which in turn triggers ERK1/2. A number of compounds have been approved for various diseases including GBM due to numerous negative clinical trials [76]. 

The first EGFR-targeted small molecule to be licensed is gefitinib; early clinical trials showed it to be safe for carcinoma patients. Nonetheless, the reaction was noted in advanced non-small-cell lung cancer (NSCLC) that was unresponsive to treatment. The EFGR gene’s particular mutation revealed how mutations interact with ATP and gefitinib with EGFR. Gefitinib’s phase II trial for recurrent GBM failed to demonstrate improved overall survival. These results cannot be achieved in GBM because EGFR mutations occur in the extracellular domain of GBM, whereas they are typically observed in the kinase domain of lung cancer. As a result, GBMs are not sensitive to first-generation EGFR inhibitors, in contrast to NSCLC. Erlotinib (Tarceva, OSI-774) was shown to prolong patients with enhanced survival rates in NSCLC [77]. Laptinib is another inhibitor with limited activity in recurrent GBM, either alone or in combination with pazapanib. In one clinical trial for recurrent GBM, afatatinib exhibited poor efficacy as a single treatment. The fundamental disadvantage of small-molecule inhibitors is their low penetrance [78]. Liu et al. [79] found that erlotinib may be disseminated throughout an intracranial U87 xenograft. Gefitinib tissue concentration was two- to threefold that of the plasma concentration in another trial, which did not result in insufficient efficacy. Tesevatinib is a second-generation RTKI that is currently being studied in GBM patients. This trial is looking into the drug’s activity in EGFRvIII-positive and -negative GBM with or without EGFR amplification. Dacomitinib, another second-generation EGFR inhibitor, was utilized in phase II trials in recurrent GBM and greatly improved certain patients [80]. Osimertinib, a third-generation EGFR inhibitor, recently demonstrated that it suppresses the constitutive activity of EGFRvIII tyrosine kinase with significant potency and inhibits its downstream signaling. There have been eight clinical studies with glioma and vandetanib, a second-generation EGFR inhibitor, to date; they all evaluated vandetanib’s effect in conjunction with other therapies such as radiotherapy and therapeutic drugs, but the results were unsatisfactory. Cetuximab is a monoclonal antibody that efficiently targets the EGFR L2 domain, limiting dimerization and cross-activation and thereby interfering with downstream signal transduction [81]. It has proven to be beneficial in treating colorectal, head, and neck cancer. The medicine was well tolerated in the progressing high-grade glioma (HGG) patient population; however, it had limited activity and failed to demonstrate benefit. In photo-immunoconjugate nanoparticles (PIC-NP), photosensitizers in cancer cells were dramatically boosted, as was light-activated cytotoxicity in U87 cells overexpressing EGFR [82].

### 3.2. PI3K/AKT/mTOR Pathway

PAM signaling affects intracellular activity via RTK and G protein-coupled receptors, including cell cycle, metabolism, migration, and death. The phosphatidylinositol 3-kinase (PI3K) phosphorylates the phosphatidylinositol 3’ hydroxyl group, resulting in the production of secondary messengers that recruit cytoplasmic proteins to the membrane. Among these are various small GTPase activity modulators, TEC family tyrosine kinase, and AGC protein kinase family AKT [82]. AKT signaling activates the serine-threonine kinase mTOR, a translational and protein synthesis regulator. PAM signaling maintains several markers of cancer cells, and both genetic and epigenetic components of this pathway are heavily influenced by the development of cancer in the central nervous system (CNS). This includes gain-of-function mutation and amplification in genes that encode the RTKs including epidermal growth factor receptor (EGFR), loss-of-function mutation of the phosphate and tensin homolog deleted on the chromosome 10 (PTEN) tumor-suppressor gene, and a mutation in several PI3K isoforms that leads to the activation of this pathway. The aberrant signaling of PAM favors the key steps of cell invasion and metastasis in a CNS tumor [83]. The implication of aberrant PAM signaling in EMT and angiogenesis is under investigation. Therefore, understanding the role of PAM pathway components is important when considering these components as a potential drug target to inhibit the often-fatal events of cell invasion and metastasis. In GBM, the PAM pathway plays a crucial role in the events of angiogenesis and expression of VEGF in cells; hence, there is a need to develop potential and effective drug molecules that inhibit the PI3K enzymes [78]. Several PI3K inhibitors are currently available, such as SF1126 (an RGDS-conjugated LY294002 prodrug), PX-866, and the dual inhibitor NVP-BEZ235. These molecules significantly suppress the expression of VEGF, thus reducing the invasion capabilities and angiogenesis of GBM cells. Among them, the inhibitor PX-866 has entered into phase II trials with patients with GBM; unfortunately, the results of these trials are low response rates. On the other hand, the combined inhibition of VEGF and VEGFR is currently used as an effective treatment to control the growth of GBM [84]. The inhibitors, such as bevacizumab, are in phase III trials. Furthermore, aflibercept is a dual inhibitor targeting the placental growth factor; however, the long-term treatment induced an invasive phenotype in GBM. In addition, RTK inhibitors like cediranib are used as effective molecules that significantly increase the overall survival rate of the mice bearing a GBM xenograft [85]. However, antiangiogenic medicines have a smaller effect than expected. This may be due to the fact that highly vascularized tissues, such as lung and brain tumors, typically multiply around existing vessels and hijack them, a phenomenon known as vessel co-option. These preexisting blood vessels avoid the need to produce new tumor vasculature, which explains why antiproliferative medications are ineffective in GBM [85].

Autophagy is an evolutionary conserved process that maintains cellular biosynthesis through protein and organelle destruction and recycling to enable metabolism and survival under famine [86]. This pathway has been linked to angiogenesis induction in numerous cancers. Autophagy has been shown in a few studies to hinder angiogenesis and enhance tumor cell growth. Autophagy is induced by many cellular stress-mediated processes. The mTOR complex 1 is a negative regulator of autophagy that inhibits the PI3K/AKT pathway. Anticancer medicines that target this system, such as the dual inhibitor NVP-BEZ235, can induce autophagy, which has both cytoprotective and antiangiogenic properties. High-grade gliomas have been shown to have lower levels of autophagy-related protein expression than low-grade gliomas. In 90% of GBM, the PI3/AKT/mTOR signaling network is shown, which acts as a prosurvival factor; hence, it is recognized as a potential drug target for combination therapy [87]. 

There are two different inhibitors that are used to block the signaling at different points of the cascade, such as GDC-0941 and rapamycin. Of these, GDC-0941 acts on apical PI3K, whereas rapamycin blocks the side arm of the network that is essential for the regulation of mTOR complex 1. These findings suggest that the PI3K networks play a crucial role in cell-specific function in GBM and should be carefully considered when incorporating the PI3K-mediated signals into complex combination therapies [88]. 

### 3.3. HGFR/c-MET

The signals from c-Met significantly promote the malignant promotion and formation of tumors, including gliomas, via an autocrine/paracrine mechanism activated by the overexpression of c-Met and its ligand hepatocyte growth factor (HGF). A number of studies reported that the activation of c-Met significantly enhances the tumor resistance to DNA damage and also enhances the tumor initiation capacity of transformed cells; these properties are recognized in the neoplastic stem cell phenotype (86). The human MET proto-oncogene is present on chromosome 7q31, and HGF is located on chromosome 7q21.1; the evidence was reported that MET plays a crucial role in glioma cell biology functions such as proliferation, growth, migration, invasion, and stemness [89]. The TCGA data analysis also suggested that 30% of GBM has the overexpression of both MET and HGF, and the autocrine HGF activation occurs in the patient population. In some studies, it was reported that MET expression was found in cytoplasm and cell membrane using immunohistochemical staining, and strong MET expression was reported in tumor cells, blood vessels, and glioma sample tissues. One study was carried out to emphasize the genetic alteration in GBM with or without IDH1 mutations, with TCGA data, and it was reported that 25 genes, of which 21 showed genetic alteration, were located on 7q31–34 [90,91,92]. Further investigation also showed that the MET gene at 7q31.2 is responsible for 47% of primary and 44% of secondary GBM; the study suggests that genetic alteration plays a role in pathogenesis. In addition, the alteration in MET is also important for progression of low-grade to secondary GBM. Also, the gain function of MET diffuses astrocytomas. The immunohistochemical staining correlated with WHO grade II survival glioma cells. In GBM, the high expression of MET was weak to moderate, and the staining intensity was noticed in 23% of the unamplified GBM and was firm only in unamplified GBM [93]. The paracrine HGF secretion from neurons greatly facilitated glioma cells’ growth and development and chemotactic invasion in MET-positive cells. In addition, HGF also acts as a chemokine and is responsible for infiltration in malignant gliomas; all of these mechanisms play a key role in the aggressive progression of GBM [94].

To understand the mechanism and role of MET signaling in GBM, the mutation in the MET signaling pathway must first be studied. According to an in vivo investigation, MET amplification is one of the most prominent oncogenic events in transgenic mouse models of GBM development [90]. Furthermore, 4% of the clinical specimens showed amplification of MET, resulting in this kinase’s overexpression and activation. The autoactivation of the METΔ7–8 mutation is a unique one, with exon 7 and 8 deletion that was found in 6% of high-grade gliomas. An RNA-seq investigation of 272 gliomas found fusion transcripts of the MET genes triggering mutations in other genes, including PTPRZ1-MET. Other fusion genes, including TFG-MET and CLIP2-MET, were also deleted in pediatric GBM. This fusion activates mutant mitogen-activated protein kinase (MAPK) signaling, which is associated with cell-cycle regulation [95].

### 3.4. NF-κB Signaling in Glioblastoma

Nuclear factor-κB (NF-κB) is a transcription factor that plays a crucial role in regulating the number of genes mediating a number of cellular processes including cell proliferation, growth, motility, survival, and cell differentiation [96]. The aberrant activation of NF-κB is a typical occurrence in a variety of cancers, including GBM, the most common and fatal type of brain tumor. In general, GBM is characterized by therapy resistance and almost unavoidable recurrence after surgery and treatment. Abnormal activation of NF-κB is responsible for a number of stimuli in GBM, and its activity has been linked to cancer stemlike cells, stimulation, cell invasion, and radiotherapy resistance [97]. The NF-κB signaling is important for several mechanisms that are activated by various stimuli. NF-κB is a dimeric DNA-binding complex composed of five family members, such as p50, p52, RelA, RelB, and c-Rel; these molecules are heterodimers, but a homodimer of RelA has also been also observed. The dimer form of NF-κB is expressed in many cells in which it is usually present in the inactive stage with the complex of specific NF-κB inhibitors until the reception of activating signals occurs. IκB is one of the most essential NF-κB inhibitors, preventing DNA binding and transcriptional regulation [98].

There are two types of NF-κB pathways: canonical and noncanonical. In the canonical NF-κB pathway, the IκB kinase (IKK/IKBK) complex leads to the phosphorylation of IκBα by IKKβ, the ubiquitination and proteasomal degradation of IκB, the dissociation of p50:RelA dimers from IκB, and the translocation of NF-κB to the nucleus. NF-κB is often activated upon stimulation of surface receptors such as tumor necrosis factor alpha (TNF) receptor 1 or interleukin-1 receptor [99]. 

NF-κB activation requires NF-κB-inducing kinase (NIK) and p52-containing NF-κB dimers in a noncanonical route. The phosphorylation of the p52 precursor form, p100, leads to its proteolytic cleavage, forming p52: RelB NF-κB dimers that translocate to the nucleus downstream of various receptors, including B-cell-activating factor receptor. In another method, the IKK complex, which is associated with DNA double-strand breaks and reactive oxygen species, can activate NF-κB. In this process, NF-κB is activated by two distinct pathways, depending on whether they are dependent or independent of IIK activity. Stressful events, such as genotoxic stress, can cause the translocation of NEMO protein to the nucleus, where they are ubiquitinated via a mechanism dependent on the ataxia and telangiectasia mutated (ATM) kinase. This causes NEMO to be exported from the nucleus to the cytoplasm, where it activates IKK and induces NF-κB. In the case of IKK-independent mechanisms, p50-containing dimers play a key role, and other kinases including casein kinase-II are required to dissociate NF-κB dimers from IκB and translocate them to the nucleus [100].

NF-κB activation has been found in various cancers, as has NF-κB downregulation in oncogene via the encouragement of tumor growth and invasion, the suppression of programmed cell death, and resistance therapy. Mutation, deregulation of NF-κB genes, or disruption of the mechanism controlling NF-κB dimer activation can all result in abnormal NF-κB activity in a variety of cancers. The most common feature of GBM is abnormal NF-κB activation, and numerous mechanisms have been linked to the downregulation of NF-κB signaling in gliomas [101]. In GBM, for example, EGFR and PDGFR are frequently abnormally active, which has been linked to the activation of numerous pathways. Oncogenic EGFR and PDGFR signaling mechanisms are important in tumor cell proliferation and invasion in GBM, although NF-κB is involved in just a small portion of these receptors’ boosting function. Furthermore, the loss of tumor suppressors such as phosphatase and tensin homolog (PTEN) and neurofibromin 1 (NF1) is linked to aberrant NF-κB activation in GBM, resulting in enhanced PI3-kinase activity. Another tumor suppressor, Krueppel-like factor 6 (KLF6), which works as a negative NF-κB regulator and contributes to NF-κB activation in GBM, is lost [102]. These findings highlight the role of the faulty NF-κB pathway in multiple stages of GBM pathogenesis and uncover several pathways upstream and downstream of NF-κB signaling. According to current research, patient-derived cells (produced from the culture of stemlike cells from individual patients’ brain tumors) demonstrate stemlike cell behavior in vitro and share the potential to generate offspring cultures with normal neural stem cells [103].

### 3.5. Wnt Pathway

The wingless/integrated (Wnt) signaling pathway is crucial in shaping essential cellular processes throughout the central nervous system’s developmental stages. It effectively regulates stem cell migration, differentiation, and self-renewal in the fetal ventricular zone. It has been amply proven that WNT signaling hyperactivation is connected with promoting malignant transformation and the development of brain tumors. The high level of -catenin expression is strongly associated with neural stem cell growth, showing its role in self-renewal [104].

Wnt signaling operates via numerous signal transduction pathways that carry messages from outside the cell into the cytoplasm and subsequently into the nucleus via cell surface receptors. There are three distinct WNT signaling pathways: the conventional WNT pathway, the noncanonical planner cell polarity pathway, and the noncanonical WNT/calcium pathway. Among these, the conventional WNT/β-catenin signaling pathway effectively governs neural stem cell (NSC) cell proliferation and fate decisions [105]. The conventional WNT response is governed by the β-catenin transcriptional coactivator, which contributes to the expression level in cells employed for regulation. For example, mature hippocampal progenitor cells are not exposed to WNT ligands, and cytoplasmic β-catenin connects with a multiprotein “destruction complex.” Adenomatous polyposis coli (APS), axis inhibition protein 1 and 2 (AXIN1/2), casein kinase 1 (CK1), and glycogen synthase kinase 3 beta (GSK3β) are all found in this complex. Both APC and AXIN1/2 play important roles in the proximity of CK1 and GSK3β, as well as the phosphorylation of β-catenin by CK1 and GSK3β, its ubiquitination by the β-transducin-repeat-containing protein (β-TrCP), and its destruction via the proteosome [106]. In the absence of WNT, members of the T-cell factor and lymphoid enhancer factor families of sequence-specific transcription factors linked to WNT-responsive DNA elements (WREs) recruit transcription factor complexes coupled to WREs. WNT ligands bind to cell-surface Frizzled (FZD)/low-density lipoprotein receptor-related protein (LRP) 5 or 6 receptor complexes. This binding contact causes AXIN1/2 to be recruited to the plasma membrane by interacting with disheveled proteins (DVLs) and the destruction complex to be inactivated [107]. The WNT pathway acts as the main signaling transduction pathway, along with other pathways such as Notch, RAS/RAF/MAPK, Hedgehog, and PI3K/Akt/mTOR, to confer stemness in GBM. The WNT pathway is frequently hyperactive in GBM tumors, allowing cells to replicate the embryonic process that leads to the critical characteristics such as proliferation and invasiveness seen in GBM tumors [108]. 

WNT5A, a noncanonical WNT molecule, enhances neuronal differentiation and is highly expressed in cell proliferation. ShRNA knockdown in GBM cells such as GBM-05 and U87MG results in a considerable reduction in cell growth. On the other hand, activation of the noncanonical WNT signaling pathway is strongly linked to GBM cell invasiveness. The expression of noncanonical factors such as WNT5A and FZD-2 also highly influence cell invasion in GBM and also highly influence prognosis; the high-level expression of WNT5A can be used to define the GBM subtypes in the TCGA dataset [109]. Furthermore, WNT5A positivity has been found in mesenchymal human GBM tissues, and it binds to tyrosine kinase-like orphan receptor (ROR) 1 or 2 and FZD, resulting in receptor internalization and the start of the PCP pathway. JNK is a PCP pathway downstream component that stimulates the WNT5A-induced production of lamellipodia and reorientation of the microtubule-organizing center (MTOC). WNT5A downstream investigations revealed a decrease in migration during healing, implying that WNT5A has a strong influence on GBM cell motility [110].

Both canonical and noncanonical routes successfully enhance components of epithelial-to-mesenchymal transition (EMT), a complex process that is essential for metastasis. EMT-activating transcription factors are receiving more attention in order to highlight the role of classical EMT in cancer biology. Although metastasis is a rare occurrence in GBM, brain cancer is classified as a grade IV glioma due to its invasive nature. However, WNT signaling is important in the activation of EMT in many tumor types; activation of the WNT/β-catenin pathway stimulates the overexpression of EMT-TFs such as Snail, Slug, and Twist [111]. AXIN, WTX, APC, and TCF4 mutations were found to be strongly related to abnormal Wnt signaling in numerous cancer types. Colorectal cancer has the most detailed characterization of these mutations. Furthermore, mutations in the WNT pathway’s key components dramatically regulate the WNT pathway. According to one study, the miR-138-2-3p and miR-770-5p are widely expressed in GBM and play an important role in Wnt signaling abnormalities in GBM, but they can also affect -catenin expression in cancers such as hepatocellular carcinoma and laryngeal tumors [112]. 

Long noncoding RNAs (lncRNA) also play a role in Wnt activation in GBM, with the expression of HOX transcript antisense RNA (HOTAIR), maternally expressed gene 3, and nuclear-enriched abundant transcript 1 (NEAT1) being negatively associated, whereas the antagonizing non-protein-coding RNA (DANCR) is positively associated. Though various publications have examined the molecularly targeted therapies that have entered clinical trials, information on their efficacy is limited. Wnt proteins bind to the Frizzled and low-density lipoprotein receptor-related protein 5/6 receptors in the canonical Wnt signaling pathway. Monoclonal antibodies such as ipafricept and vantictumab, which strongly block the Wnt signaling pathway, disturb this relationship. Vantictumab, for example, has been shown in a phase I trial to be well tolerated with no side effects [113].

### 3.6. Notch Pathway Deregulation in Brain Tumors and Brain CSCs

The aberration of the Notch pathway is observed in many tumor types; the downregulation of Notch pathway is associated with the development of a number of diseases via both germline and somatic mutation [114]. Mutation in the Notch pathway leads to malignancy, and it was observed in several solid tumors, including a brain tumor. Gliomas and medulloblastomas are the most common types of brain tumors; several studies reported that the aberration of Notch components in brain tumors, for example, the overexpression of Notch1, 3 and 4, ASCL1, and Hey1, are highly correlated with a higher grade of glioma and poor prognosis. These results indicated that the activation of Notch signaling effectively promotes the undifferentiated and aggressive tumor phenotype. The concept of cancer stem cell (CSC) theory says that CSCs play a crucial role in cell proliferation development and are the most important reason for metastasis and relapse [115]. Brain CSCs can grow as neurospheres in serum-free medium and share similarities with NSCs, significantly increasing the expression of stem cell markers including GFAP, CD133, and nestin, and can differentiate into all three neural lineages, since Notch signaling acts an effective regulator of the NSC to maintain the NSC tumor counterpart [116,117]. For example, the expression of stem cell markers in brain CSCs is effectively regulated by intracellular modulators such as Numb4 and Numb4 Δ7, which act as inhibitors and activators for Notch signaling, respectively. Brain CSCs are more resistant under hypoxic conditions, and the depletion of HIF-1α influences the proliferation of glioma-derived brain CSCs by blocking the interaction of HIF-1α and NICD. These data suggest that Notch singling is essential for maintaining the characteristic of stemness and tumorigenic potential of brain CSCs; hence, it acts as a potential drug target for Notch-based therapies [118]. 

In GBM, both mRNA and protein levels are more highly expressed than normal brain cells. The elevated levels of VEGF and pAKT and reduced levels of PTEN also correlate with brain tumor cells [119]. For instance, the overexpression of Notch1 was observed in >1-year-survival patients compared with <1 year. This study suggested the controversial role of Notch in GBM. The overexpression of Notch significantly correlates with higher-grade and primary tumors. Notch2 overexpression in GBM influences stemness genes such as nestin and SOX2, astrocyte destiny genes vimentin and GFAP, and antiapoptotic protein, but it is inversely connected to the expression of Olig2, CNP, and PLP1 as well as proapoptotic proteins BAX and BCLAF1 [117]. Hey1 overexpression is linked to tumor grade and survival due to Notch signaling dysfunction. Several studies, however, found low levels of Notch1, 2, p300, and MAML1 expression in GBM [115]. Notch-related genes are significantly enriched in pSTAT3 patients in the mesenchymal subtype. Verhaak et al. [120] found that Notch signaling is overexpressed in the classical subtype, whereas Hey2 and Dll3 expression is low in perineural GBM with high Notch expression. Even though about 30% of GBMs contain IDH mutations, the majority of GBMs in the proneural subtype have IDH mutations and a proneural gene expression pattern. According to Spino et al. [121], IDH mutants are typically low-grade and have high Dll3 expression. The noncanonical Notch pathway is also important in gliomagenesis. Huber et al. discovered that GBMs have higher levels of Deltex1 (DTX1) expression than normal brain cells and that many pathways, including RTK/PI3K/PKB and the antiapoptotic protein Mcl-1, are implicated in glioma aggressiveness [117].

Reversibility is one of the peculiar characteristic features of epigenetic alteration that makes it a potential and effective drug target with which to explore the abnormalities of the cancer epigenome. Until now, there have not been many studies that give details about the epigenetic regulation of Notch signaling in GBM. The study carried out by Tsung et al. reported the methylation status of the Hey1 factor and its contribution to GBM pathogenesis, and they found low levels of CpG methylation when compared with a healthy brain owing to the overexpression of Hey1 [118]. Another study found that inducing apoptosis in GBM using sodium butyrate, a histone deacetylase inhibitor, drastically reduced Hey1 expression while increasing DNMT1 levels. Hey1 knockdown effectively reduced cell migration and proliferation [122]. 

MicroRNAs (miRNAs) are short noncoding RNA molecules of a small size that hold significant importance in the regulation of genes. They achieve this by binding to the messenger RNAs (mRNAs) of protein-coding genes, suppressing translation, or promoting mRNA degradation and instability. Regulating the gene expression and the number of cellular processes is associated with mRNA alterations. Sun et al. and others reported the contribution of 32 mRNAs in the Notch signaling pathway, and 6 of them play a crucial role in GBM and the Notch regulatory network (Figure 2) [123,124]. The miR-34 family is one of the most studied miRNAs which downregulate in GBM tissues more than in normal brain tissue and are overexpressed in wild-type p53 GBM more than in mutant p53 GBM. The miRNAs such as miR-34a and miR-34a-5p act as tumor-suppressive factors that significantly inhibit cell proliferation and cell-cycle progression by targeting the expression of Notch1, -2, c-Met, and CDK6. Wu et al. reported that the low levels of miR-34c-3p and miR-34c-5p are associated with a high grade of GBM [125]. 

The overexpression of both these miRNAs strongly inhibits the cell invasion of GBM and promotes S-phase arrest, enhances cell apoptosis, and reduces the expression of Notch 2 [126]. In another study, miR-148a and miR-31 were reported to be associated with poor survival in GBM. The high levels of miR-31 are associated with proliferation and immune-response genes, while the high levels of miR-148a are associated with hypoxia-induced and extracellular matrix genes. The inhibition of these miRNAs in GBM mouse models showed that downregulation of both miRNAs prolongs animal survival, suppresses tumor growth, and depletes the stem cell pool [127].

MK-0752 represents a γ-secretase inhibitor that specifically and efficiently targets Aβ40, demonstrating an IC50 value of 5 nM. In CSCs, where the treatment of MK-0752 with a chemotherapy agent against metastatic and refractive cancer occurred, studies reported that the combination of MK-0752 and tocilizumab significantly reduced the BCSC numbers and suppressed tumor growth. The combination of MK-0752 with dalotuzumab effectively inhibited cell proliferation, angiogenesis, and stem cell propagation in a phase I trial. Likewise, the combination of MK-0752 and docetaxel is used against advanced breast cancer [128].

Demcizumab (OMP-21M18) is an IgG2 monoclonal antibody that targets the Notch ligand DLL4, which plays a crucial role in the Notch signaling pathway and maintains the CSCs’ proliferation and tumor angiogenesis. Demcizumab effectively targets the Notch signaling pathway; hence, it is entering the clinical trial phase to treat solid tumors [129] Table 1 lists the key small-molecule inhibitors that are currently in use in clinical trials. 

### 3.7. Self-Renewal Pathway

Classical Hh signaling is a key route in cancer that regulates embryonic development, stem cell control, and tissue homeostasis. Sonic Hh (SHh), desert Hh (DHh), and Indian Hh (IHh) are three gene homologs in Hh that bind to PTCH and block PTCH on Smoothened (Smo) and are relieved, allowing Gli to translocate into the nucleus and trigger the transcription factor. Tumorigenesis is caused by aberrant Hh signaling expression in a variety of cancers, including breast, small-cell lung, stomach, prostate, and hematological malignancies. Several Hh inhibitors have been reported in preliminary clinical trials. Self-renewal and chemoresistance in pancreatic CSCs are efficiently suppressed by inhibiting the Hh signaling pathway. The Hh pathway also regulates BCSC self-renewal via Bmi-1. In addition, Smo inhibitor therapy reduces the expression of CSC markers and makes tumor cells more sensitive to docetaxel in the PDX model [130].

Cyclopamine is one of the bioactive compounds that inhibit the Hh signaling pathway, specifically targeting Smo. According to the findings, cyclopamine is an effective inhibitor of gemcitabine resistance in pancreatic cancer cells, greatly suppressing the expression of CSC markers. It is a potent anticancer agent against bladder tumorigenesis and inhibits bladder CSC self-renewal [131]. Vismodegib (GDC-0449) is another potent small-molecule antagonist of the Hh pathway that interacts with Smo and inhibits aberrant Hh pathway activity. The FDA approved this molecule as an effective agent against the Hh pathway for treating advanced basal cell carcinoma. Vismodegib promotes tumor cell differentiation by preventing a hair follicle-like destiny and mediating BCC regression. It is currently being utilized in clinical trials as part of a combination therapy for numerous cancers, including GBM, small-cell lung cancer, basal cell carcinoma, prostatic cancer, and pancreatic cancer. It also activates caspase 3 and induces PARP cleavage, inducing apoptosis in pancreatic CSCs. Despite having very high anticancer activities against BCC, clinical trials have revealed severe side effects, limited selectivity for CSCs, and drug resistance. As a result, new approaches are required [132]. 

GANT61 is the most extensively used antagonist that targets Gli proteins and effectively blocks Hh signaling. It has strong inhibitory activity and highly influences the cell proliferation and migration of cancer cells including pancreatic, breast, and other malignancies. It also inhibits the growth of paclitaxel-resistant cells and significantly reduces the proportion of CSCs in TNBC cells. Also, the combination of GANT61 with mTOR inhibition provides an effective practical approach for inhibiting pancreatic CSCs, enhancing pancreatic cancer treatment [133]. 

### 3.8. STAT3

STAT3 is activated by conventional phosphorylation of Tyr-705 at its carboxyl terminus. STAT3 post-translational modification causes phosphorylation at Ser-727, acetylation, and methylation, which may all work together to modulate its function. STAT3 overexpression leads to dimerization via reciprocal interactions with Src homology 2 protein domains that bind phosphotyrosine. The homo- and heterodimers translocate to the nucleus and interact with STAT-binding proteins via specific consensus sequences known as interferon gamma-activated sequences (GAS) and DNA-response elements of numerous genes such as c-Jun, Bcl-2, myc, p21, and HIF-1a. STAT3 transcriptional activity is dependent on the sequence specificity of nuclear STAT-binding proteins, which increases the activation of intracellular serine/threonine kinases, which regulate STAT3 phosphorylation at Ser-727 [134]. PKC and mTOR have been demonstrated to phosphorylate STAT3-Ser727, influencing its activity and preventing phosphorylation at Tyr-705. Most critically, Ser727 influences energy consumption and cell respiration via regulating mitochondrial function. Along with ATP generation, STAT3-Ser-727 phosphorylation activates the electron transport chain and mitochondrial membrane polarization. It then modulates cytochrome c oxidase activity and ROS generation. Upstream signaling proteins such as the epidermal growth factor receptor (EGFR), the fibroblast growth factor receptor (FGFR), the insulin growth factor-1 receptor (IFG-IR), and the interleukin-6 receptor (IL-6R) all play an important role in STAT-mediated transcription [135,136]. 

Several variables are successfully involved in regulating the STAT3 signaling cascade and operate as critical checkpoints of cell proliferation under normal physiological settings. The suppressor of cytokine signaling (SOCS) protein is a JAK negative regulator that inhibits JAK/STAT signaling. Other proteins, including SH2-protein tyrosine phosphatases and T-cell PTP, dephosphorylate and inactivate the phosphorylated-STAT3 dimers [137]. 

Furthermore, the functional control of STAT3 involves activating a stable dimer formation through SH2 acetylation at lysine K685, whereas the methylation of K140 in STAT3 serves as a mechanism to negatively regulate transcriptional activation. Furthermore, the interaction of E3 SUMO protein ligase-activated STAT3 and STAT3-interacting protein (StIPI) suppresses gene transcription by interfering with STAT3 protein’s DNA-binding ability. STAT3 activation increases the invasive characteristics of GBM through the interaction of VEGF and hypoxia-inducible factor 1 (HIF-1). Under the hypoxia condition, STAT3 nuclear translocation leads to the overexpression of VEGF and enhances the endothelial tube formation in gliomas [134]. This report suggests that STAT3 is one of the potential and effective targets for treating tumor neovascularization. The anti-VEGF therapy also reported that the overexpression of STAT3 in glioma patients and STAT3 inhibitors could be therefore used to enhance the anticancer treatment. In addition, the activation of STAT3 induces the transcription of proinvasive factors such as matric mettaloproteinase-9 and -2, fascin-1, and focal adhesion kinase (FAK) in GBM. It also induces the transcription of miR-182-5p, which suppresses the expression of protocadherin-8 (PCDH8) signaling and promotes the invasion and migration of GBM cells. A study reported that the constitutive activation of STAT3 promotes the accumulation of M2 tumor-associated macrophages (TAMs), which play a key role in suppressing the antitumor mechanism and inducing tolerance to tumor antigens. The expression of TAMs is associated with a high grade of GBM [93]. 

## 4. Interaction of Glioma Stem Cells with Surrounding Cells

Over the last decade, the understanding of biology has emphasized the important role of stem cells in the maintenance of multicellular organisms and how they play a crucial role in the growth and development of tumors. Cancer stem cells have three basic characteristics: differentiation, self-renewal, and the potential for effective proliferation. The presence of CSCs in GBM was demonstrated via specific antigenic markers. CSCs can form new neurospheres that support their stemlike properties. CD133 is one of the stem cell markers that have been found on cell surfaces that contribute to cell growth. There are 100 CD133+ cells that have been discovered that play an important role in the development of cancers with similar histological features. CD133+ cells are more involved in the development of new neurospheres than CD133- cells. The higher the expression of CD133+ cells, the more severe the malignancy. A few studies, on the other hand, found that CD133- in human GBM biopsies, which were later stereotactically transplanted in mice brains, resulted in the development of both CD133+ and CD133- cells [138]. 

Normal stem cells have morphological and functional anatomical niches that are critical for self-renewal. According to a new study, cancer stem cells in brain tumors have a perivascular niche that connects normal neural progenitors with the vasculature. CSCs promote the establishment of their perivascular niche by secreting a variety of proangiogenic factors, including VEGF [139]. Oxygen tension is one of the elements that has a strong influence on the normal physiology of cells and serves as a key signal for development, with low oxygen tension related to the maintenance of an undifferentiated cell state. Hypoxia increases self-renewal while inhibiting neural cell differentiation. The study carried out with a mouse model reported that hypoxia acts as a functional component of the normal stem cell niche, though it is essential to the maintenance of stemness in cancer cells. Cellular differentiation is the basic characteristic feature of cells that are unidirectional and irreversible, but several studies reported the potential plasticity of cellular differentiation [140]. With the help of several defined molecules such as c-Myc, Oct4, Sox2, and Klf4, several fully differentiated cells have been reprogrammed into pluripotent states. Specifically, c-Myc, Sox2, and Oct4 play a critical role in the self-renewal of brain stem cells. in some cases, cancer stem cells also show overlap of transcriptional circuitry with embryonic stem cells. When compared to normal cells, cancer cells have greater plasticity. Owing to this, they can change their phenotypes based on the surrounding environmental conditions. Several studies reported that restricted oxygen conditions enhance the fraction of cells positive for a CSC marker in established cancer cells and lead to the expression of stem cell markers [141]. In hypoxia conditions, CSCs exhibit distinct gene expression patterns and an elevated level of VEGF expression. CSCs effectively regulate the target proteins including HIF2A and transcriptional targets like Glut1 and Serpin B9 under hypoxic environmental conditions. They also regulate the expression of HIF1α via post-translational modification; it was reported that HIF2α shows more complexity with the regulation of both transcriptional and post-translational levels. At different oxygen levels, the expression level of HIF protein also shows different effects. The high expression of HIF2α was recorded under 5% oxygen levels, whereas HIF1α expression was recorded in both cancer stem cells and nonstem cells at severe hypoxic conditions [139]. 

Basically, there are three major niches associated with GSCs: the perivascular niche, the hypoxic niche, and the immune niche. The perivascular niche is surrounded by a definite number of blood vessels that are essential for tumor growth. It produces a bigger number of CSCs in hypoxic conditions, indicating that it is a positive participant in the maintenance of CSCs by supporting the essential stem cells for multipotency, tumorigenicity, and self-renewal. Both HIF1α and HIF2α seem to overlap, with 75% homologies. In addition, the aggressiveness of a tumor is associated with a hypoxic response, and GSCs show a large number of immune cells in the GBM immune niche [142].

This hypoxic state promotes HIF1α secretion via T-cell receptors, resulting in increased lytic ability of CD8+ T lymphocytes and IF-γ secretion via CD4+ cells, inflammatory cytokine production, and proliferation. Furthermore, it recruits immunosuppressive cells via GSC signaling, including TAMs, which significantly promotes angiogenesis and inhibits the immune response by secreting chemokines and growth factors β1, SDF-1, and soluble colony-stimulating factor-1 (sCSF-1) and expressing surface molecules that engross inhibitory molecules on effector immune cells, resulting in the formation of an immunosuppressive microenvironment. Hence, targeting any one of the niches, particularly the perivascular, immune, and hypoxic niches, is a crucial strategy to overcome resistance and is an alternative therapy for GBM [143].

## 5. Glioma Stem Cells as a Target for Treatment: Current Target-Based Therapy

Several reports were published that emphasize the therapeutic strategies and current therapy in use to conquer GBM. However, there is a need to understand the mechanism of action of currently used drugs against targets of GBM. Several papers discussed in this review emphasize the comprehensive targeting of kinases (such as PI3K, FAK, DYRK, and 3-phosphoinositide-dependent kinase 1 (PDK1)) via practical design approaches involving heterocyclic compounds such as triazines, pyrimidines, indoles, oxoindoles, 6:5 fused heterocycles, and other variations [144]. Another factor that has been effectively employed for developing agents against GBM is histone deacetylase (HDAC). The drug design approaches elucidated in this review distinctly illustrate the adaptability of the three-component model for inhibiting HDAC. In particular, modifying the surface recognition segment of the pharmacophore responsible for HDAC inhibition has garnered significant attention from medicinal chemists aiming to harness anti-GBM effects by targeting various HDAC isoforms. Numerous studies exploring the structure–activity relationship (SAR) have been conducted to craft inhibitors for isocitrate dehydrogenase (IDH) and translocator protein (TSPO). Importantly, extensively investigating optimized scaffolds for inhibiting IDH and TSPO involved thoroughly exploring modifications at multiple locations, aiming to better understand how such changes influence their activity [145].

Given the favorable results, these studies are expected to serve as models for various future initiatives in developing IDH and TSPO inhibitors as possible medicines against GBM. Proteins such as protein disulfide isomerase (PDI), tubulin, and hypoxia-inducible factor are also involved. During the drug development phase, over 98% of tiny drug compounds are unable to cross the blood–brain barrier (BBB), which operates as a barrier for over 95% of drug molecules. Over 98% of small therapeutic compounds face a barrier in traversing the blood–brain barrier (BBB), with over 95% of drug molecules being impeded during the drug development phase. Consequently, pharmaceutical companies do not heavily prioritize targeted drug delivery to the brain. Earlier research has additionally demonstrated that the BBB is a dynamic interface, displaying its structure and function alterations during specific pathological circumstances [146].

Given these formidable obstacles, GBM cells can exhibit an intense capacity to infiltrate adjacent tissues and advance rapidly. Individual GBM cells can aggressively generate tumors as they infiltrate the neighboring tissues, eventually overcoming the formidable blood–brain barrier (BBB) through a multistep progression. GBM cells move and concentrate around existing blood arteries, causing astrocytic endfoot processes to be displaced from these channels. Notably, the participation of TGF-2, caveolin-1, reactive oxygen species (ROS), and proinflammatory peptides in inducing matrix metalloproteinase (MMP)-mediated tight junction degradation considerably contributes to GBM cell BBB breach. Cancer stem cells (CSCs) constitute a subset of cells present within a tumor mass that are responsible for generating tumors and propelling malignant advancement even post-treatment. Substantial empirical and clinical substantiation points toward the capability of CSCs to withstand the effects of ionizing radiation and chemotherapy [147]. CSCs exhibit various cellular attributes contributing to their resistance to chemotherapy and radiation, including heightened DNA damage repair abilities, amplified survival signaling pathways, and heightened expression of reactive oxygen species (ROS) scavengers. It is worth highlighting that resistance to TMZ, a challenging problem, is primarily orchestrated by glioma stem cells (GSCs). Insights into this matter suggest that radiation and chemotherapy protocols prompt the generation of enriched groups of stemlike CD133+ cells through heightened DNA repair mechanisms. Recent investigations identified consistent markers for GSCs, encompassing CD133, CD44, CD15, CD70, S100A4, ALDH1A3, Nanog, SOX-2, and nestin. The findings from fate-mapping investigations utilizing genetic barcoding demonstrated that chemotherapy exerts evolutionary selective pressure, resulting in the proliferation of GSCs resistant to drugs. Despite GSCs constituting a small fraction of GBM tumors, their capacity to drive the resurgence of tumor diversity positions them as prospective subjects for novel anticancer therapeutic strategies [148].

Recently, the mitochondrial metabolic pathway was targeted for drug repurposing okagainst GBM treatment. The Warburg effect, one of the well-known metabolic phenomena of cancer cells, restricts both oxidative and mitochondrial metabolism and enhances the anaerobic metabolism. However, it was reported that the mitochondrial function plays a crucial role in GSC survival and the maintenance of their stemlike properties [118]. The study also reported that the effect of mitochondrial inhibitors of GSCs, such as oligomycin A and antimycin A, significantly reduce the cell viability about 100-fold more than TMZ. However, these inhibitors are used as potential inhibitors and have not been evaluated in humans. In the large-scale screening of clinically tested compounds and FDA-approved drugs, there are three inhibitors, that is, trifluoperazine, mitoxantrone, and pyrvinium pamoate, that show significant mitochondrial inhibitory properties, and these drugs effectively decrease the GSC viability about 50-fold more than TMZ. The study conducted by Aponte et al. [137] reported that neurotransmitter signal pathway inhibitors can suppress the proliferation of GSCs and their survival. The authors reported that trihexyphenidyl is an effective acetylcholine receptor antagonist used for treating Parkinson’s disease and rizatriptan significantly kills GSCs. Two categories of alkylating compounds are frequently employed in clinical settings for GBM therapy: TMZ and nitrosoureas. MGMT stands as a significant DNA repair enzyme that plays a role in the resistance of GBM to TMZ.

The correlation between survival improvement and the epigenetic silencing of the MGMT gene in GBM has been established. Furthermore, this correlation with outcomes has been observed regardless of the chosen treatment approach, whether it is chemotherapy or radiotherapy. Recent research indicates that the MGMT status lacks predictive significance in primary GBMs. Initially, Eramo et al. [149] explored the chemoresistance of GSCs, followed by Beier et al. [26] who observed that TMZ could preferentially reduce clonogenic and tumorigenic cells in a dose-responsive fashion with minimal impact on overall viability. The authors concluded that cells exhibiting stem cell-like characteristics were preferentially diminished regardless of their CD133 or MGMT status. Cheng et al. [150] demonstrated that CD133+ cells displayed notably reduced viability in response to TMZ treatment as opposed to CD133− tumor cells. Furthermore, CD133+ cells with methylated MGMT exhibited characteristics reminiscent of stem cells. In a separate study, Pistollato et al. [151] revealed heightened resistance in central, hypoxic CD133 CSCs when compared to peripheral counterparts, owing to heightened MGMT expression.

Another study demonstrated that the absence of CD133 expression in secondary GBMs originating from lower-grade gliomas. They proposed that secondary GBMs often exhibit IDH1 mutations, which could potentially hinder the in vitro growth of GBM cells. The methylation of MGMT is prevalent in secondary GBMs and can serve as a predictive indicator for TMZ therapy. Given these findings, we can reasonably deduce that GSCs exhibiting elevated CD133 levels and resistance to TMZ are indicative of primary GBMs, originating directly from NSCs. Conversely, tumors exhibiting reduced CD133 expression and responding positively to TMZ treatment are likely to be secondary GBMs, stemming from lower-grade astrocytomas [152].

## 6. Recent Advancements in Metabolic Reprogramming of GBM and GSCs

Metabolic reprogramming in GBM and GSCs has also been observed, with substantial activation of the glycolytic pathway, which includes genes from the aldehyde dehydrogenase (ALDH) family [153]. ALDH is made up of detoxifying enzymes that are in charge of converting (retin)aldehydes into retinoids [154]. In a xenograft model, human mammary cells with enhanced ALDH activity showed the greatest lineage differentiation potential and the best growth capability, demonstrating that the ALDH1 positive cell population is enriched in mammary stem cells. Furthermore, repeated passages revealed that the ALDH1-positive population had a stronger tumorigenic capacity than the ALDH-negative population [155]. The exact isoform of ALDH responsible for the enzymatic activity remains controversial; however, ALDH1A3 is considered to have a predominant role in GSCs [156]. Intake of necessary nutrients is a critical phenomenon in all live cells for maintaining cellular homeostasis through a variety of metabolic pathways and energy production. Energy is obtained in normal cells by glucose metabolism, which converts glucose to pyruvate and then enters the tricarboxylic acid cycle (TCA) to make ATP molecules via oxidative phosphorylation. GBM and GSCs have increased glycolytic flux for faster ATP molecule generation due to their rapid growth and food restriction; this process guarantees that GSCs meet their energy production requirements even under tough hypoxic environments [157]. Several studies have been undertaken to assess metabolism-related genes and their impact on cancer patients, particularly those with GBM. Understanding the molecular phenotypes of GBM and the biology of GSCs requires molecular insights into complicated connections between the metabolism and molecular abnormalities [158,159]. 

Gliomas stand alone as a type of cancer where the presence of the IDH1/2 mutation serves as a distinct and favorable prognostic indicator. The typical enzymatic pathway assisted by wild-type IDH1 and IDH2 involves the oxidative decarboxylation of isocitrate to -ketoglutarate (α-KG) as well as the production of the reduced form of nicotinamide adenine dinucleotide phosphate (NADPH). The IDH mutation, on the other hand, causes the use of α-KG and NADPH to produce an oncometabolite, D-2-hydroxyglutarate (D2HG), which has a direct role in creating the G-CIMP phenotype and producing aberrant DNA methylation in gliomas. D2HG has also been found to be a competitive inhibitor of certain -ketoglutarate-dependent dioxygenases. As a result, the promethylation effects of the IDH mutation may be linked to the levels of expression of the impacted dioxygenases in a specific cell [160]. Lai et al. [161] demonstrated that GBMs carrying the IDH mutation could potentially emerge from neural cells on a committed lineage, whereas gliomas lacking the IDH mutation might arise from neural stem cells (NSCs). IDH1 mutation is notably prevalent in secondary GBMs stemming from lower-grade astrocytoma. Consequently, tumors bearing the IDH1 mutation, lacking CD133 expression, and showing MGMT positivity correspond to secondary GBMs. On the other hand, tumors marked by CD133 expression and lacking IDH1 mutation align with primary GBMs. The presence of mutant IDH1 in these tumors triggers pronounced hypoxia, fostering the proliferation of GSCs and ultimately contributing to therapy resistance. Additionally, it has been documented that D2HG inhibits the activity of branched-chain amino acid aminotransferase (BCAT) transaminases, making glioma cells reliant on glutamine and more susceptible to glutaminase inhibition. Notably, both primary GBM and other gliomas with wild-type IDH exhibit a similar dependence on glutamine, suggesting distinct contributors to the glutamine-dependent energy pathway in gliomas. Furthermore, the metabolic imbalances linked to IDH mutations may also impact glycolysis, TCA cycle metabolism, and the production of phospholipids and reactive oxygen species (ROS). Recent clinical trials identified innovative inhibitors designed to target both IDH1 and IDH2. This breakthrough can potentially manage the malignancies carrying IDH mutations, such as gliomas, by mechanistically reducing 2-HG levels and subsequently curbing invasiveness. Notably, Novartis’s IDH305 focuses on IDH1 mutations, and ongoing trials in gliomas and other cancers with IDH1 R132 mutations are active (NCT02381886). Furthermore, both Forma Therapeutics’ FT2102 and Bayer’s BAY1436032 target IDH1 R132 tumors (R132X for BAY1436032), with trials actively enrolling patients across several tumor types, including GBM (NCT03684811 and NCT02746081) [162]. 

In drug-resistant GBM, the lipid metabolism pathway undergoes considerable alterations. Lactate was also discovered to be a major alternate energy and biosynthetic substrate for oxidative GSCs, sustaining their proliferation in the absence of glucose [163]. Lipids have an important role in cell membrane construction, energy storage, and signaling. Lipid metabolism is disrupted in GBM, with increased fatty acid production and overexpression of lipid transporters such CD36 and fatty acid binding protein (FABP). These changes contribute to drug resistance by increasing cell survival and increasing the supply of lipid precursors for membrane production and energy generation [12]. These metabolic changes aid in the development of treatment resistance in GBM. TMZ is a frequently used chemotherapeutic drug for treating GBM, a particularly aggressive brain tumor. Nonetheless, the emergence of TMZ resistance remains a major barrier in GBM treatment. A recent study suggested that alterations in lipid metabolism, such as the synthesis of steroids and arachidonate metabolites, may contribute to the development of TMZ resistance in GBM.

Analytical technology advancements, such as imaging mass spectrometry, have aided in the spatial mapping of lipid metabolism in GBM and GSCs. This breakthrough enables the identification of lipid distributions and prospective targets for lipid-based therapy. A detailed examination of a GBM patient-derived xenograft (PDX) model, which included mass spectrometry imaging, histology, magnetic resonance imaging, phosphoproteomics, and mRNA sequencing, revealed that the distribution of the EGFR inhibitor erlotinib within intracranial tumors was insufficient to inhibit EGFR tyrosine kinase signaling, despite promising efficacy in vitro studies [164,165].

## 7. Limitations of Current Glioblastoma Treatments

Glioblastoma, known as GBM, is a highly aggressive brain disorder that poses significant challenges in treatment due to its limited survival rates. The necessity for targeted therapeutic strategies is evident to improve treatment effectiveness and extend the survival of individuals affected by GBM. Utilizing siRNA therapy holds promise as a potential avenue for addressing GBM. TMZ stands as a potent chemotherapy drug predominantly utilized in the treatment of GBM. The metabolites of TMZ, specifically MTIC, exert direct depletion effects on the O6 methyl guanine bases within DNA, resulting in DNA damage and the alteration of regulatory mechanisms. Prolonged use of TMZ leads to resistance in malignant glioma cells, undermining the effectiveness of these medications. Consequently, the supplementary administration of the aforementioned agents exhibits the potential to counteract TMZ resistance in malignant gliomas, enhancing patient survival rates and overall outcomes. As a result, further investigations are imperative to gain deeper insights into the GBM environment [166].

The use of siRNA as a potent inhibitor for targeted therapy proved successful in suppressing genes or signaling pathways associated with GBM. Nevertheless, employing siRNA therapy for GBM faces multiple challenges, including issues of immunogenicity, limited cellular absorption, insufficient bloodstream distribution, fragile stability in blood, and constrained ability to breach the blood–brain barrier. The utilization of angiopep-2 (An2)-modified exosomes loaded with STST3siRNA (Exo-An2-siRNA) represents a promising therapeutic candidate to elevate the efficacy of GBM treatment. These Exo-An2–siRNA complexes exhibited a remarkable stability within the bloodstream, an improved cellular absorption, and a notable capability to penetrate the blood–brain barrier. Furthermore, Exo-An2–siRNA demonstrated heightened anti-GBM efficacy in vitro, underscoring the potential of exosomes to shield siRNA while effectively targeting GBM through An2 modification. Creating an appropriate siRNA drug delivery system could serve as an efficacious therapeutic strategy against GBM by circumventing the blood–brain barrier, bolstering delivery to counter the formidable nature of GBM, and ultimately leading to enhanced survival rates for patients affected by this condition. Precision therapy and immunotherapy represent potential treatments that could offer increased effectiveness, tolerability, and promise in addressing the challenges of treating undruggable GBM [167]. Ongoing efforts involve immunotherapy techniques targeting immune checkpoint inhibitors like PD-1/PD-L1 and CTLA-4 for treating GBM. Advancing clinical studies are pivotal in enhancing the efficacy of immunotherapy for GBM [168].

## 8. Concluding Remarks and Future Perspectives

Understanding the molecular mechanism and treatment of GBM presents a multifaceted and challenging task that is not an unsolvable problem. The large number of genomic data and other research approaches enable scientists to explore relevant hypotheses grounded in the trends observed in genomics across populations. Numerous medications currently employed in clinical settings have shown the capability to interact with signaling pathways associated with GBM. A substantial portion of these drugs are undergoing repurposing efforts to benefit GBM patients, potentially progressing through clinical trial stages. Concurrently, some drugs exhibit potential modes of action against GBM and are undergoing preliminary investigations in preclinical settings. As GSCs have emerged as significant contributors to postsurgery tumor relapse, recent research has focused on scrutinizing molecular pathways targeting this cell subset. Genomic techniques unveiled a distinct molecular signature of the illness and key pathways for concentrated research. Despite the advancement in identifying more precise molecular categories of GBM, the actualization of targeted treatments for particular GBM subtypes remains unrealized. The abundance of unsuccessful clinical trials implies that combining therapies is likely the most encouraging avenue for GBM treatment, placing significant focus on optimizing drug design and pharmacokinetic characteristics. Hence, delving into the possibilities offered by combined treatment approaches offers substantial potential. Integrating stem cells with conventional therapies might present an improved outlook for prognoses.

## Figures and Tables

**Figure 1 cancers-16-00102-f001:**
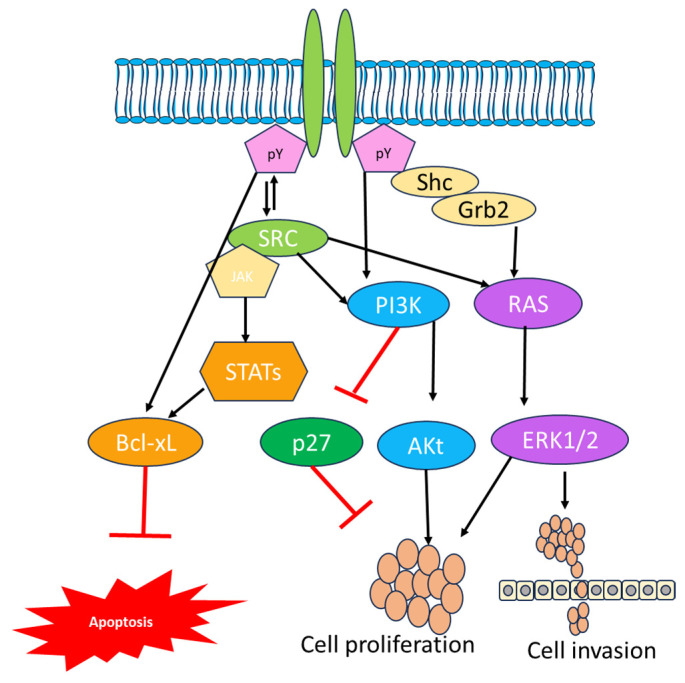
Representation of the EGFR signaling pathway in cancer, illustrating the involvement of genes for signal transferring from the extracellular environment to the nucleus through SRC and Grb2 growth factor receptor-bound protein.

**Figure 2 cancers-16-00102-f002:**
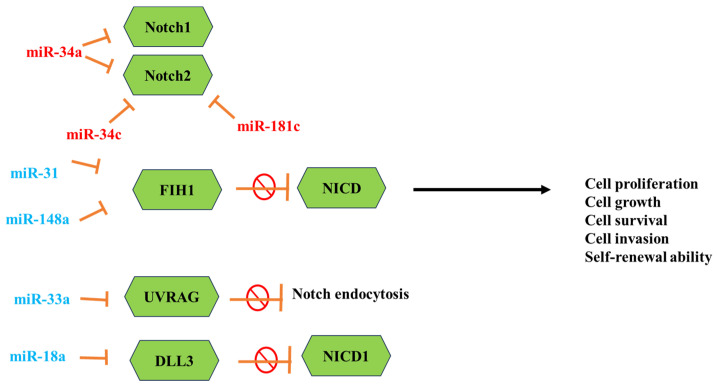
Representation of the important miRNAs that effectively regulate the Notch signaling pathway in glioblastoma. Red miRNAs are downregulated, and cyan ones are upregulated.

**Table 1 cancers-16-00102-t001:** Small-molecule inhibitors in clinical trials for glioblastoma and other cancers.

S.NO	Name of the Drug	Drug Target	Clinical Trial	Indication
1	Ventictumab	FZD1, 2, and 5	Phase I	Solid tumors, breast, and pancreatic
2	Ipafrecept	FZD8	Phase I	Pancreatic, hepatocellular, ovarian
3	Cirmtuzumab		Phase II	Breast, lymphoma cancer
4	WNT974	ROR1	Phase II	Head and neck, colorectal, and pancreatic
5	PRI-724	Β-catenin/CBP	Phase II	Solid tumors
6	ETC-59	PORCN	Phase I	Solid tumors
7	CGX-1321	PORCN	Phase I	Solid tumors
8	Crizotinib	TKI, HGFR	Phase I	GBM
9	Imatinib	BCR-ABL, c-KIT	Phase II	GBM
10	Acalabrutinib (ACP-196)	TKI	Phase II	GBM

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
