# Peer review of "Progress in Glioma Stem Cell Research"

_cancers, 2023, doi:10.3390/cancers16010102_

Round 1

Reviewer 1 Report

Comments and Suggestions for Authors

The manuscript by Ramar et al. aims to review the current molecular targets for GBM therapy. Overall, the manuscript is well-written, providing a complete overview of the main issues associated with GBM treatment. Figures and tables are clear, too. I have no concerns about this article and I recommend it for acceptance in the present form.

Author Response

We are grateful for the reviewer’s valuable comments.

Reviewer 2 Report

Comments and Suggestions for Authors

Although this review raises and discusses several interesting topics, it lacks focus and clarity, and the topic mentioned in the title is not addressed in a focused way. This manuscript requires a very substantial general revision before it is further considered for publication. I do not think it is currently up to the standard of Cancers, a high-impact high quality journal.

Comments on the Quality of English Language

An extensive general language review is required. For example, in the abstract, note how the sentence "In this review we emphasize the molecular mechanisms of GBM, can acquire resistance to treatment" is poorly constructed. In the title, maybe "Glioma Stem Cell" should be "Glioma Stem Cells". In yet another example, "Molecular Mechanism of Glioma Cells" is a poorly chosen subtitle.

Author Response

Reviewer #2: Although this review raises and discusses several interesting topics, it lacks focus and clarity, and the topic mentioned in the title is not addressed in a focused way. This manuscript requires a very substantial general revision before it is further considered for publication. I do not think it is currently up to the standard of Cancers, a high-impact high quality journal.

Response: Thank you for highlighting those points.  We have implemented significant changes based on the reviewer’s suggestions, which encompassed altering the title, revising the abstract, incorporating additional glioma stem cell (GSC) related information, and placing a stronger emphasis on GSC, especially recent advancements in metabolic reprogramming within glioblastoma (GBM) and GSC.

We have emphasized the areas where substantial changes were made in red, though we didn’t specifically highlight the grammatical errors.

An extensive general language review is required. For example, in the abstract, note how the sentence "In this review we emphasize the molecular mechanisms of GBM, can acquire resistance to treatment" is poorly constructed. In the title, maybe "Glioma Stem Cell" should be "Glioma Stem Cells". In yet another example, "Molecular Mechanism of Glioma Cells" is a poorly chosen subtitle.An extensive general language review is required. For example, in the abstract, note how the sentence "In this review we emphasize the molecular mechanisms of GBM, can acquire resistance to treatment" is poorly constructed. In the title, maybe "Glioma Stem Cell" should be "Glioma Stem Cells". In yet another example, "Molecular Mechanism of Glioma Cells" is a poorly chosen subtitle.

Response:  We greatly appreciate the reviewer’s critical comments. We have incorporated changes as per the reviewer’s suggestions. Furthermore, we have rectified numerous previous grammatical errors. To maintain simplicity, we chose not to highlight these, focusing solely on emphasizing significant changes in red.

Reviewer 3 Report

Comments and Suggestions for Authors

In this manuscript, Ramar and colleagues summarized molecular mechanism of GBM, interaction between GBM and microenvironment, and advances in GBM therapy.  Importantly, the authors offered insights into critical issues and future research directions in the field of GBM. In general, the manuscript is overall balanced and well-written.

If possible, please summarize more recent developments and latest advances in metabolic reprogramming in GBM.

I, or probably many readers, would like to read more about the advances in specific and effective targeting of GBM.

There are a number of grammatical/editorial mistakes throughout the manuscript, which make it challenging to read at times. Below is a non-exhaustive list of sentences that remain to be amended: “In this review we emphasize the molecular mechanisms of GBM, can acquire resistance to treatment.”; “5.Glioma Stem Cells as a Target for Treatment Current Target-Based Therapy”.

A thorough review of this manuscript for grammar and readability would be of benefit.

Comments on the Quality of English Language

Minor editing of English language required.

Author Response

Reviewer #3: In this manuscript, Ramar and colleagues summarized molecular mechanism of GBM, interaction between GBM and microenvironment, and advances in GBM therapy.  Importantly, the authors offered insights into critical issues and future research directions in the field of GBM. In general, the manuscript is overall balanced and well-written.

If possible, please summarize more recent developments and latest advances in metabolic reprogramming in GBM.

I, or probably many readers, would like to read more about the advances in specific and effective targeting of GBM.

Response: We are grateful for the reviewer’s valuable comments. As per the reviewer’s suggestions, we have included a new section titles “Recent Advancements in Metabolic Reprogramming of GBM and GSCs”, and highlighted in red.

There are a number of grammatical/editorial mistakes throughout the manuscript, which make it challenging to read at times. Below is a non-exhaustive list of sentences that remain to be amended: “In this review we emphasize the molecular mechanisms of GBM, can acquire resistance to treatment.”; “5.Glioma Stem Cells as a Target for Treatment Current Target-Based Therapy”.

A thorough review of this manuscript for grammar and readability would be of benefit.

Response: We are grateful for the reviewer’s valuable comments. We have addressed the multitude of grammatical errors, including those highlighted by the reviewer. To uphold simplicity, we chose not to emphasize these corrections; instead, we highlighted only the significant changes in scientific information in red.

Round 2

Reviewer 2 Report

Comments and Suggestions for Authors

This manuscript was considerably improved after revision. My criticisms were addressed.